# Therapeutic Drug Monitoring (TDM) Implementation in Public Hospitals in Greece in 2003 and 2021: A Comparative Analysis of TDM Evolution over the Years

**DOI:** 10.3390/pharmaceutics15092181

**Published:** 2023-08-23

**Authors:** Gavriela Voulgaridou, Theodora Paraskeva, Georgia Ragia, Natalia Atzemian, Konstantina Portokallidou, George Kolios, Konstantinos Arvanitidis, Vangelis G. Manolopoulos

**Affiliations:** 1Laboratory of Pharmacology, Medical School, Democritus University of Thrace, 68100 Alexandroupolis, Greece; gavoulga@affil.duth.gr (G.V.); thparask@affil.duth.gr (T.P.); gragia@med.duth.gr (G.R.); ntatzemi@med.duth.gr (N.A.); konporto@affil.duth.gr (K.P.); gkolios@med.duth.gr (G.K.); karvanit@med.duth.gr (K.A.); 2IMPReS—Individualised Medicine & Pharmacological Research Solutions Center, 68100 Alexandroupolis, Greece; 3Clinical Pharmacology and Pharmacogenetics Unit, Academic General Hospital of Alexandroupolis, 68100 Alexandroupolis, Greece

**Keywords:** therapeutic drug monitoring, hospitals, pharmacogenomics, digoxin, antiepileptics, antibiotics, immunosuppressants, psychiatric drugs, substances of abuse, Greece

## Abstract

Therapeutic drug monitoring (TDM) is the clinical practice of measuring drug concentrations. TDM can be used to determine treatment efficacy and to prevent the occurrence or reduce the risk of drug-induced side effects, being, thus, a tool of personalized medicine. Drugs for which TDM is applied should have a narrow therapeutic range and exhibit both significant pharmacokinetic variability and a predefined target concentration range. The aim of our study was to assess the current status of TDM in Greek public hospitals and estimate its progress over the last 20 years. All Greek public hospitals were contacted to provide data and details on the clinical uptake of TDM in Greece for the years 2003 and 2021 through a structured questionnaire. Data from 113 out of 132 Greek hospitals were collected in 2003, whereas for 2021, we have collected data from 98 out of 122 hospitals. Among these, in 2003 and 2021, 64 and 51 hospitals, respectively, performed TDM. Antiepileptics and antibiotics were the most common drug categories monitored in both years. The total number of drug measurement assays decreased from 2003 to 2021 (153,313 ± 7794 vs. 90,065 ± 5698; *p* = 0.043). In direct comparisons between hospitals where TDM was performed both in 2003 and 2021 (n = 35), the mean number of measurements was found to decrease for most drugs, including carbamazepine (198.8 ± 46.6 vs. 46.6 ± 10.1, *p* < 0.001), phenytoin (253.6 ± 59 vs. 120 ± 34.3; *p* = 0.001), amikacin (147.3 ± 65.2 vs. 91.1 ± 71.4; *p* = 0.033), digoxin (783.2 ± 226.70 vs. 165.9 ± 28.9; *p* < 0.001), and theophylline (71.5 ± 28.7 vs. 11.9 ± 6.4; *p* = 0.004). Only for vancomycin, a significant increase in measurements was recorded (206.1 ± 96.1 vs. 789.1 ± 282.8; *p* = 0.012). In conclusion, our findings show that TDM clinical implementation is losing ground in Greek hospitals. Efforts and initiatives to reverse this trend are urgently needed.

## 1. Introduction

In 1538, Paracelsus said, “All substances are poisons; there is none that is not a poison. The right dose differentiates a poison from a remedy”. This demonstrates that knowledge that the right dosage of a drug was very important has existed since the early years of medicine. In 1964, Finney published one of the first structured reviews about therapeutic drug monitoring [1]. TDM is an approach for personalizing pharmacotherapy by measuring drug concentrations in patient fluids, mainly in plasma or whole blood. There is a steady need for continuous measurement of medications in almost all fields of medicine, especially in neurology, psychiatry [2], cardiology [3], oncology [4], and transplantation medicine [5,6].

TDM can be performed to determine the effectiveness of treatment [7] as it can prevent the occurrence or reduce the risk of adverse drug reactions (ADRs), including, importantly, severe toxicity [7,8]. It can confirm to clinicians that the prescribed doses are safe and effective. Thus, disturbing situations can be avoided where the optimal therapeutic doses required for treatment are commonly determined after ADR occurrence [9]. The large inter-individual variability in the efficacy of different drugs makes it difficult to individualize dosing [2]. These pharmacokinetic variations may be due to treatment resistance, potential drug–drug interactions, genetic variations in drug metabolism, and various clinical and non-clinical conditions such as pregnancy and obesity [2,10]. In addition, TDM enables the assessment of compliance to drug therapy and individualization of potential pharmacokinetic specificities [8], facilitating, therefore, the prescription of the appropriate drug dosage regimens, bringing maximum clinical benefit from pharmacotherapy and reducing the risk of under-treatment or other adverse effects.

To be eligible for TDM, a drug should satisfy one or more of three main criteria: (a) a narrow therapeutic range, (b) significant pharmacokinetic variability, and (c) a predefined target concentration range [7]. In clinical routine, TDM is applied for approximately 20 commonly used drugs, including antiepileptics (e.g., phenytoin, carbamazepine, phenobarbital, primidone, valproic acid, clonazepam), immunosuppressants (e.g., cyclosporine, tacrolimus, mycophenolate, sirolimus), certain antibiotics (e.g., vancomycin, gentamicin, amikacin), cardiac drugs (e.g., digoxin, procainamide, lidocaine), respiratory drugs (e.g., theophylline, caffeine), antineoplastics (e.g., methotrexate) and drugs used in psychiatry (e.g., lithium, tricyclic antidepressants) [11].

The majority of laboratories implement TDM by use of immunoassay analyzers and immune-based methods, such as enzyme-linked immunosorbent assays (ELISAs) and chemiluminescent immunoassays (CLIA). Liquid chromatography (LC)–mass spectrometer (MS) is also used in TDM [12,13]. LC–MS/MS methods are considered the gold standards for TDM, as they have high specificity; however, high running costs limit their routine application in hospitals [13,14]. It should be noted, however, that all these highly specific techniques, except for their high cost, require lengthy preparation time and properly trained personnel [12]. Instead of these techniques, new-generation biosensor analysis techniques can be used, offering direct analysis at a reduced cost; however, they are still far from being routinely applied in a clinical setting [12,15].

Although, occasionally, contrary opinions have been expressed about TDM [16], its importance in achieving personalized medicine in routine clinical practice is undoubted. Currently, there is a dearth of evidence regarding the uptake and implementation of TDM and clinical toxicological measurements in hospitals worldwide and also on how this uptake has evolved through the years. In this study, we aim to address this issue by collecting and comparing data on TDM in Greek hospitals by means of a comprehensive questionnaire sent to all hospitals in 2004 (requesting data for 2003) and again in 2022 (requesting data for 2021).

## 2. Materials and Methods

### 2.1. Study Design

This was a series, cross-sectional study based on a questionnaire that was sent in 2004 and 2022 to all Greek public hospitals. More specifically, in 2004, the questionnaire was sent by email or fax, together with a letter from the scientific supervisor of the study explaining the aims of the survey. Data were collected from January to March 2004 describing TDM implementation in 2003. In 2022, an online survey was conducted using the same questionnaire designed in Google Forms. The questionnaire link was sent via email along with a letter explaining the scope of the survey and instructions for completing the questionnaire to all public hospitals in Greece. This study was conducted from March to November 2022 and covered data for 2021. The study progress flowchart and the response rate are shown in Figure 1.

Ethical approval and permission for the 2021 study were obtained from the Research Ethical Committee of the Democritus University of Thrace [No. 2041/20-02-2022]. For the 2003 study, no ethical approval was deemed necessary at the time by the Ethics Committee. Data from the Academic General Hospital of Alexandroupolis were provided from the Laboratory of Pharmacology, Medical School, Democritus University of Thrace, where the Clinical Pharmacology and Pharmacogenomics Unit of our hospital is located.

### 2.2. Questionnaire

The questionnaire was designed for the specific aims of this project by two experts in the field of TDM. The initial questionnaire was reviewed for face and content validity to ensure that the content was easily understood, accurate, free of grammatical and syntax errors, and avoiding repetition and bias. Then, the tool was piloted to a sample of 8 hospitals to test its reliability. The questionnaire was further improved, according to initial results.

The final questionnaire used consisted of 3 sections. Section one covered questions about the hospital (name, hospital category, and city), the person completing the questionnaire (specialty), and the director or head of the laboratory performing the pharmacological measurements. The second section comprised questions on the availability of TDM in the hospital; which drugs were measured, how many measurements were performed for each drug in the previous year, at which specific laboratories/clinics within the hospital these measurements were performed, how many people work in these laboratories, what specialty and training they have, and what equipment was used for TDM implementation. In hospitals where TDM was not applied, information on TDM solutions when physicians requested drug measurements was requested. The third section included one question about the application of toxicological measurements in the hospitals (which substances of abuse were measured, and how many analyses were performed for each of these in the previous year). In the questionnaire used in 2022, a fourth section was added, comprising 3 questions about pharmacogenomics. Specifically, this section focused on the performance of pharmacogenomic analyses in hospitals and which genes were tested. The questionnaire was written and distributed in the Greek language.

### 2.3. Statistical Analysis

Free text responses were coded and sorted where possible. Both descriptive and inferential statistics were applied. In descriptive statistics, categorical variables were presented as percentages or as numbers, and continuous variables as mean ± standard deviation (SD). The Kolmogorov–Smirnov test was performed to check the assumptions of normality for continuous variables to choose the appropriate parametric or non-parametric tests. Wilcoxon test was performed to compare the drug measurement between 2003 and 2021 in hospitals that performed TDM in both years. McNemar test was applied to compare the number of hospitals that measured drugs in both years. Statistical analysis was performed with SPSS Statistics v.23.0 (IBM Corp., Armonk, NY, USA). The statistical significance level was set at <0.05.

## 3. Results

In 2003, the questionnaire was sent to all public Greek hospitals (n = 132); 113 hospitals (85.6%) responded to our survey. In 2021, a total of 122 hospitals were identified and reached, with 98 of them responding (80.3% response rate). Due to consolidation and other reasons, the number of public hospitals in Greece in 2021 was reduced by 10; among these 10 hospitals, 6 performed TDM, and 1 performed toxicological tests in 2003, while the remaining 4 hospitals had not responded to our survey. In both years, one hospital did not provide a response on the toxicological analysis performance; therefore, the total number included in the analysis is one less. Table 1 shows the total number of hospitals that performed TDM and toxicological analyses.

### 3.1. Drug Monitoring

Fifty-one hospitals responded that they conducted TDM in 2021; however, one hospital did not specify the type of drugs analyzed. Therefore, 50 hospitals were included in further drug analyses. Table 2 compares the number of hospitals that performed measurements for each drug in 2003 and 2021. A total of 33 drugs were measured; 11 of these drugs were measured only in 2003, while three drugs were measured only in 2021. As a result, for these drugs, the McNemar test was not applied. The number of hospitals that measured phenytoin (40 vs. 28; *p* = 0.050), phenobarbital (34 vs. 20; *p* = 0.016), digoxin (52 vs. 37; *p* = 0.038), and theophylline (26 vs. 7; *p* < 0.001) significantly decreased from 2003 to 2021.

The most commonly monitored drug category was antiepileptics. Forty-two hospitals in 2003 and thirty-nine hospitals in 2021 carried out TDM for antiepileptics. Phenytoin was the most frequently measured drug (95.2%) of this class in 2003, while in 2021, valproic acid was the only drug that remained at a similar level of measurement as in 2003 (92.4% in 2003 vs. 92.3% in 2021; Figure 2a). In both 2003 and 2021, 22 hospitals measured antibiotics. Vancomycin was the most commonly measured antibiotic in both years (95.5%), followed by amikacin which was measured by 72.7% of hospitals in 2003. However, in 2021, this percentage decreased by approximately 50% (36.4%) (Figure 2e). For psychiatric drugs, 20 hospitals in 2003 and 16 in 2021 performed TDM. Lithium, followed by benzodiazepines and tricyclic antidepressants, were measured in 2003 by approximately 70%, 40%, and 30% of hospitals, respectively. In 2021, lithium measurement increased (81.3%), while benzodiazepines and tricyclic acids decreased considerably (18.8% and 12.5%, respectively; Figure 2b). Figure 2 depicts the frequency of hospitals for each category of measured drugs.

The total number of drug measurements performed per year for each drug and their mean values (±SD) are presented in Table 3. For the majority of drugs, there was a significant decrease in 2021 compared to 2003. Digoxin was the drug with the highest number of measurements in 2003, followed by valproic acid. For both drugs, a decrease in measurements is present in time (83.2% decrease for digoxin, 27.8% decrease for valproic acid). In contrast, measurements of six drugs were increased in 2021. Vancomycin determinations increased by 71.1%, followed by gentamycin (38.5%) and acetaminophen (34.4%), while everolimus, sirolimus, and levetiracetam TDM entered clinical practice after 2003.

A total of 153,313 drug measurements were performed in Greek hospitals in 2003 and 90,065 in 2021 (corresponding approximately to a 41% reduction in country-wide-performed TDM determinations). In 2003, 32.1% of the total number was for antiepileptic drugs, followed by immunosuppressants (17.3%), antibiotics (13.5%), psychiatric drugs (4.5%), and analgesics (2.6%). In 2021, the top category of drugs measured were antibiotics (43.8%), followed by antiepileptic drugs (23.6%) and immunosuppressants (19.8%). The percentages of psychiatric drugs and analgesics for which TDM was still performed were very small (2.8% and 2.4%, respectively) (Figure 3). The mean value of the total number of drug measurements decreased from 5110 (±7901) in 2003 to 4093 (±6453) in 2021, corresponding to a reduction of 19.2%.

#### Hospitals Performing TDM in Both 2003 and 2021

Thirty-five hospitals were performing TDM in both 2003 and 2021. The Wilcoxon test shows a significant decrease in carbamazepine (198.8 ± 46.6 vs. 46.6 ± 10.1; *p* < 0.001), phenytoin (253.6 ± 59 vs. 120 ± 34.3; *p* = 0.001), tobramycin (16 ± 7 vs. 0.57 ± 0.6; *p* = 0.017), amikacin (147.3 ± 65.2 vs. 91.1 ± 71.4; *p* = 0.033), digoxin (783.2 ± 226.70 vs. 165.9 ± 28.9; *p* < 0.001), theophylline (71.5 ± 28.7 vs. 11.9 ± 6.4; *p* = 0.004) measurements in 2021, while only vancomycin measurements were significantly higher (206.1 ± 96.1 vs. 789.1 ± 282.8; *p* = 0.012) in 2021 compared to 2003 (Table 4). A significant decrease (21%) was observed in drug measurements that were performed in 2003, compared to the drug measurements performed in 2021 (2907.2 ± 552.8 vs. 2309.3 ± 524; *p* = 0.043) in all Greek public hospitals (Figure 4).

### 3.2. Toxicological Analysis for Substances of Abuse

A total of 9 out of 112 hospitals reported that they performed toxicological analyses of substances of abuse in 2003, and 6 out of 97 hospitals in 2021. One of the nine hospitals performing toxicological analysis in 2003 did not exist in 2021. Opioids, cocaine, and tetrahydrocannabinol (THC) were the substances measured at all hospitals in both years. Amphetamines and benzodiazepines were also measured by all hospitals in 2021, in contrast to 2003, when only six hospitals measured them. Buprenorphine and phencyclidine were measured only in 2021 by one hospital (Figure 5).

### 3.3. Characteristics of TDM Implementation

The type of laboratory in which TDM was performed was mostly the biochemistry unit of the hospitals at 37.2% in 2003 and 40.8% in 2021. The number of microbiology units performing TDM decreased in 2021 (n = 4) compared to 2003 (n = 13). Only a few hospitals performed their TDM analyses in dedicated in-house clinical pharmacology laboratories. Twenty and seventeen hospitals in 2003 and 2021, respectively, sent samples for TDM to an external lab. A total of 242 and 260 persons were employed in the laboratories of 64 and 51 hospitals that performed TDM in 2003 and 2021, respectively. Their personnel were mostly lab technicians (44.6% in 2003 and 61.5% in 2021), followed by medical doctors (28.5% in 2003 and 21.5% in 2021) (Table 5).

Hospitals were also asked about the equipment they used for TDM implementation. The total number of analyzers used for TDM implementation was reduced from 89 in 2003 to 57 in 2021. Most of them were based on immunofluorescence assays both in 2003 (77.5%) and 2021 (79.1%). Gas and liquid chromatography were no longer applied, whereas the use of electrolyte analyzers decreased from 3.4% in 2003 to 1.5% in 2021 (Figure 6). The evolution of the technology was also evident. Abbott AXSYM was the most commonly used analyzer in 2003 (in 32.8% of the labs), followed by Abbott TDX (21.3%). In 2021, Abbott Architect (32.8%) and ROCHE Integra (31.3%) were the standard equipment in the majority of TDM laboratories.

### 3.4. Pharmacogenomics

Only two hospitals responded that they perform pharmacogenomic analyses in their laboratories, but only one of them provided specific data on the examined genes. This is the University General Hospital of Alexandroupolis, where pharmacogenomics analyses are applied in the Laboratory of Pharmacology of the Medical School. Pharmacogenomics tests implemented include genotyping of CYP450 isoenzymes (CYP2D6, CYP2C19, CYP1A2, CYP2C9, CYP3A5) and other drug catabolizing enzymes (DPYD), drug transporters and receptors (SLCO1B1, SLC6A4, DRD2, DRD3, 5HT2CR, 5HT2AR) and drug targets (VKORC1). However, as reported, additional polymorphisms of pharmacogenetic interest are occasionally performed in response to specific requests by hospital clinicians.

## 4. Discussion

TDM is a useful tool for precision medicine to achieve therapeutic response and reduce the risk of toxicity. According to the International Association of Therapeutic Drug Monitoring and Clinical Toxicology (IATDMCT), TDM can be used either for determining patient dosing regimens or for pharmacokinetic/pharmacodynamic monitoring [17]. In addition, TDM can enhance treatment cost-effectiveness [18,19]. However, there is insufficient data on TDM implementation in different countries worldwide. To the best of our knowledge, no study has compared TDM evolution in time in any country. In addition, this is the seminal study conducted to ever evaluate TDM implementation in Greece. It is comprehensive since the questionnaire was sent to all public Greek hospitals, and a high percentage replied both times (85% in 2004 and 80.3% in 2022, providing data for TDM for the years 2003 and 2021, respectively). A similar percentage of hospitals implemented TDM in 2003 (56.6%) compared to 2021 (52%). It should be noted that the percentage of hospitals implementing TDM in both years is relatively low. In Malaysia, in 2006, 64.5% of hospitals implemented TDM, a higher rate compared to Greece in 2003 [20].

TDM is applied to a wide range of drugs administered in various pathological conditions, such as immunosuppressive drugs used in transplantation or for the treatment of inflammatory bowel disease and autoimmune hepatitis [21], antifungal agents [22], antibiotics administered in critically ill patients [9] or in bacterial infections [23], antiepileptics [24], and anticancer drugs [4]. The main categories of drugs measured in 2003 and 2021 in Greece were antiepileptics, psychiatric drugs, analgesics, immunosuppressants, and antibiotics. Antiepileptics are measured by the majority of hospitals in Greece, although the total number of TDM measurements in this category was dramatically reduced in 2021 (21,250) vs. 49,190 in 2003. In recent decades, TDM of antiepileptic drugs has been used as a complementary method of epilepsy treatment to determine optimal therapeutic doses, as these drugs have high pharmacological variability [25,26]. Of 27 antiepileptic drugs that are globally available, only for 6 of them TDM is performed in Greece, namely carbamazepine, phenytoin, valproic acid, phenobarbital, primidone, and levetiracetam. All these drugs’ measurements decreased in 2021 compared to 2003. Carbamazepine (*p* < 0.001) and phenytoin (*p* = 0.001) showed a significant decrease in 2021 among hospitals that practiced TDM in both years (n = 35). These drugs are the older first-generation antiepileptic drugs [26]. With the exception of the “second” generation drug levetiracetam, several other new-generation antiepileptic drugs like pregabalin, tiagabine, and stiripentol [27] are not measured in Greek hospitals. Apparently, and sadly, in Greece, the reduction in requests for TDM for old antiepileptics has not been matched by an increase in demand for the new ones. This might reflect a lack of interest and/or knowledge by the clinicians that TDM is applicable and useful also for the newer antiepileptics.

Antibiotics were the second most frequently measured drug category in both 2003 and 2021. Vancomycin is the most common antibiotic measured in almost all hospitals that perform TDM for antibiotics. Other antibiotics that are measured in Greek hospitals are amikacin, gentamicin, tobramycin, and vancomycin, while netilmicin is no longer measured in 2021. Comparing the numbers of the tests performed, there was a significant increase in vancomycin measurements (*p* = 0.012) and a significant decrease in amikacin (*p* = 0.033) and tobramycin (*p* = 0.017) measurements in 2021 as compared to 2003. Vancomycin was also the most frequently measured antibiotic in German hospitals (75%), followed by meropenem and piperacillin [28], which are not measured in Greek hospitals. The increase in vancomycin measurements is mainly due to the occurrence of vancomycin-resistant *Staphylococcus aureus*, as well as vancomycin-resistant enterococcus, which have been linked to under-dosing, leading to an increased need for monitoring of vancomycin levels [29]. Thus, antibiotics had the largest overall increase in tests performed in 2021 (39,424) compared to 2003 (20,649). Despite this increase, however, a significant class of antibiotics, β-lactams, are not measured in Greek hospitals. Even though β-lactams have a wide therapeutic range, overdose has been associated with toxicity [30], while on the other hand, a large number of patients in intensive care units (ICUs) do not achieve the required therapeutic target [31]. Therefore, TDM in critically ill patients receiving β-lactams is necessary. Reduced availability for TDM implementation for β-lactams compared to other antibiotics is also observed in German hospitals [28]. A possible explanation for this is the demanding method for β-lactams measurement, which requires more complex techniques such as LC–MS/MS [32].

Both the number of tests and the number of hospitals performing TDM for theophylline measurements have significantly decreased between 2003 and 2021. Theophylline is a drug prescribed mainly for the treatment of asthma. The drug has a low therapeutic range [33] and is known to cause severe side effects [34]. The observed decrease can be explained by the fact that theophylline is an old drug that is no longer frequently prescribed, as new drugs have come out for the treatment of asthma that are more effective and have fewer side effects.

Another important finding of our study was related to digoxin. While in 2003, digoxin was the drug with the most tests performed yearly, in 2021, there was a significant drop in tests as well as in the number of hospitals offering them. Digoxin is an antiarrhythmic drug, and for many years it was administered for heart failure and atrial fibrillation treatment. It is necessary to implement TDM in patients receiving digoxin since its therapeutic range is narrow and its toxicity significant [35]. However, there has been a significant change in the way clinicians prescribe the drug. While the recent European guidelines still recommend digoxin therapy in patients with atrial fibrillation as a class I indication (level B) [36], its use for heart failure has almost been eliminated. This is confirmed by the findings of our study. Other studies have found similar reductions. For example, prescriptions decreased from 14,514 in 1991 to 7448 in 2004 per 100,000 beneficiaries (*p* < 0.001) in America [37]. However, it should be emphasized that TDM for digoxin remains important, especially for the elderly (>65 years), to prevent toxicity and achieve optimal levels [38].

Various drugs such as netilmicin, primidone, aminoglycosides, amiodarone, ethosuximide, quinidine, and procainamide are losing ground in TDM over the years, reflecting their diminished clinical use mainly due to the emergence of new drugs with better therapeutic benefits and/or fewer side effects. On the other hand, two immunosuppressive agents, everolimus and sirolimus, did not exist in 2003 and, therefore, were not measured before.

In recent years, guidelines for TDM implementation in several drug categories or specific drugs have been published. A recent consensus guideline for TDM in neuropsychopharmacology includes 154 drugs for which TDM should be performed [2]. However, only eight of them are measured in Greek hospitals. Furthermore, guidelines support the use of TDM in patients with inflammatory bowel disease receiving biological agents to monitor and guide their treatment [39]. Our study showed that in Greece, no TDM for biological agents is performed. The same is true for antiretroviral (ARV) drugs, although several studies support their monitoring [40,41,42,43].

In 2003, only nine hospitals measured substances of abuse, while in 2021, this number decreased to six. This can be partly attributed to the creation by the state of dedicated forensic laboratories performing these analyses. However, from a therapeutic perspective, toxicological testing is preferred in a healthcare setting, as these measurements provide important information for physicians to make correct diagnoses and prescriptions for their patients [44]. TDM for substances of abuse offered by public hospitals in Greece includes opiates, amphetamines, cocaine, THC, barbiturates, benzodiazepines, and ethanol. These analyses mainly require immunoassays using commercially available immunoassay kits and enzyme assays for the determination of alcohol, thus replacing the need for complex and expensive techniques such as LC/MS [45].

In 2003 and 2021, most of the hospitals performed TDM analyses in the biochemical laboratory (37.2% and 40.8%, respectively). The laboratories used may differ between countries or even between hospitals. In Malaysia, the main laboratories where TDM is applied are the pharmacy and biochemistry [20]. In Australia, the complexity of the analyses determines the type of laboratory performing it; for example, complex analyses such as cyclosporine are performed in the pharmacology laboratory, while simpler ones are performed in the biochemistry or chemical pathology laboratory [46].

As our study shows, in Greece, the staff in the TDM laboratories are mainly laboratory technologists, physicians, chemists, and biologists. However, TDM is a multidisciplinary activity. Thus, to provide the best practice of TDM, apart from the need to have well-trained staff to ensure the accuracy and correctness of the results, the collaboration of clinicians, nurses, and laboratory staff is crucial [47].

In both years assessed, immunochemical methods comprise the majority of analytical techniques used for TDM in Greek hospitals, followed by biochemical methods. This is well justified; immunoassays provide a rapid and cost-effective solution through commercially available kits that allow TDM to be applied in automated analyzers [47]. However, in recent years other techniques, such as chromatography, have started gaining ground in hospitals. The most recent technological advances in chromatography are combined with mass spectrometry (LC/MS), leading to an important new tool for more precise TDM services [48]. LC-MS/MS method allows simultaneous measurement of multiple drugs in a single sample [49] and has high specificity [50,51]. In a study investigating methods for measuring immunosuppressants, it was observed that different techniques are used depending on the drug. More specifically, the LC–MS/MS method was mainly used for sirolimus and everolimus measurements, while for tacrolimus, the preferred method was immunoassay [52]. It should be noted, however, that LC–MS/MS methods have several limitations, such as high cost, increased test time, highly trained personnel required, restricted availability of commercial calibrators, and limited standardization efforts. These limitations may explain the absence of this method for TDM performance in Greek hospitals [47,53,54]. On the other side, they consist of an area of development of more comprehensive and robust TDM assays [55].

Pharmacogenomics can be a complementary tool, along with TDM, to improve drug therapy and optimize pharmacological regimens by better explaining interindividual variability in drug response [56,57,58]. Together with TDM, they can be used as complementary tools to further improve and personalize drug therapy in various therapeutic areas, most notably antiepileptics and psychiatric drugs [8,24,59,60]. Unfortunately, despite its wide application in clinical practice, from all Greek public hospitals, only two responded positively to the application of pharmacogenomic analyses. Reasons for this lack of implementation of pharmacogenomic testing in Greece include the absence of reimbursement schemes from public and private insurance companies, insufficient knowledge of clinicians, and a lack of specific guidelines on implementation from medical specialty societies.

A major strength of this study is that it is the first worldwide to comprehensively assess and quantify the implementation of TDM in an entire country in two different years (2003 and 2021) and compare its evolution within a time span of almost twenty years. An additional strength is the high rate of responses from public hospitals in both years. On the other side, there are also some limitations. Although the health care system in Greece consists mainly of public hospitals, there are also private hospitals, some of which may be implementing TDM. However, in our study, only public Greek hospitals were included. Another limitation is that, in 2021, restrictions on patient movement and barriers to access to hospitals due to COVID-19 were still in existence. However, hospital laboratories continued performing TDM analyses on a regular basis throughout this period. Furthermore, yearly data from our laboratory show a downward trend in TDM from 2015 onwards, with the curve reaching a peak of decline in 2020 and again increasing in 2021, reaching 2019 levels (unpublished data).

## 5. Conclusions

TDM is an essential tool in daily clinical practice and is an important part of personalized medicine. Our results demonstrate that the use of ΤDM in Greece is relatively limited, and over the years, the implementation of TDM has declined. Antiepileptics and antibiotics were the most common drug categories measured in 2021. Unfortunately, TDM for new antiepileptic drugs has yet to be adopted. In addition, TDM tests for several drug categories, such as biological agents and antiretrovirals, are still not implemented in Greek hospitals. These findings demonstrate the need to increase TDM in Greece by adding new drugs following the guidelines to approach the best pharmacotherapy regimen for patients. Overall, in the current era of precision medicine, TDM remains a valid and useful tool for achieving this goal, and a refreshed emphasis on its use and its expansion to additional drugs would be beneficial for patients, doctors, and health systems alike.

## Figures and Tables

**Figure 1 pharmaceutics-15-02181-f001:**
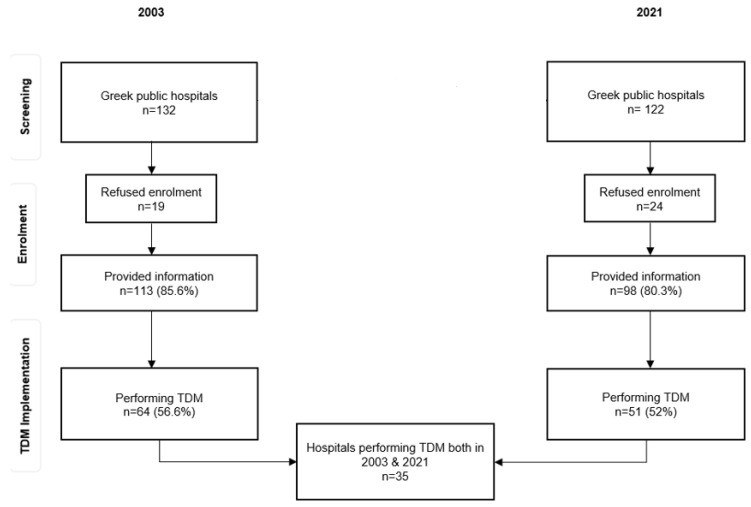
Flowchart showing the response rate of the hospitals.

**Figure 2 pharmaceutics-15-02181-f002:**
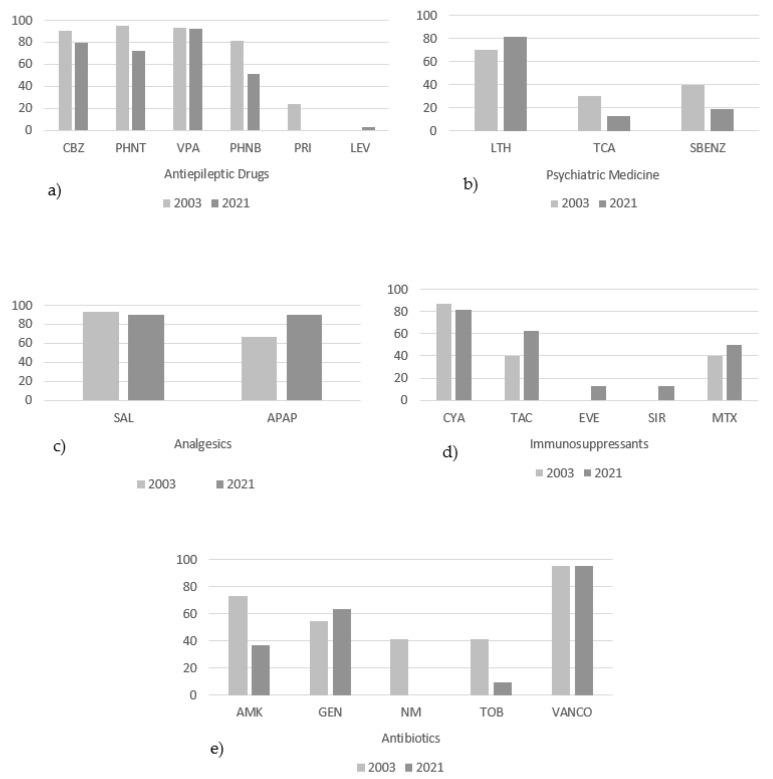
Frequency of hospitals performing TDM measurements per drug/category of drugs. (**a**) antiepileptic drug measured in 42 hospitals in 2003 and in 39 hospitals in 2021; (**b**) psychiatric drugs measured in 20 hospitals in 2003 and 16 hospitals in 2021; (**c**) analgesics measured in 15 hospitals in 2003 and 10 hospitals in 2021; (**d**) immunosuppressants measured of 15 hospitals in 2003 and 16 hospitals in 2021; (**e**) antibiotics measured of 22 hospitals in 2003 and 22 hospitals in 2021. CBZ = carbamazepine; PHNT = phenytoin; VPA = valproic acid; PHNB = phenobarbital; PRI = primidone; LEV = levetiracetam; LTH = lithium; TCA = tricyclic antidepressants; SBENZ = benzodiazepines serum; SAL = salicylate; APAP = acetaminophen; CYA = cyclosporine; TAC = tacrolimus; EVE = everolimus; SIR = sirolimus; MTX = methotrexate; AMK = amikacin; GEN = gentamycin; NM = netilmicin; TOB = tobramycin; VANCO = vancomycin.

**Figure 3 pharmaceutics-15-02181-f003:**
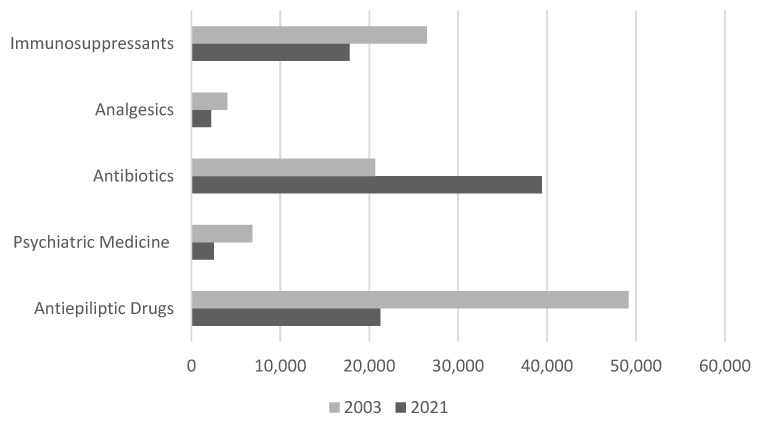
Total number of drugs per category of medicines in 2003 and in 2021.

**Figure 4 pharmaceutics-15-02181-f004:**
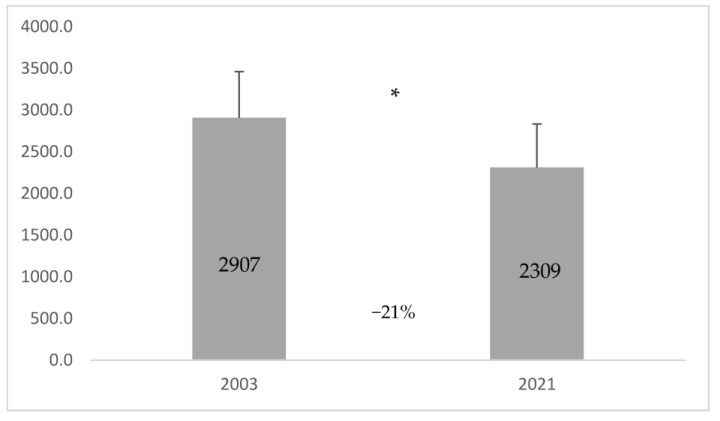
Total mean value (±standard deviation) of drug measurements in 2003 and 2021 in the hospitals which performed such measurements in both years. * Wilcoxon test: *p*-value = 0.043.

**Figure 5 pharmaceutics-15-02181-f005:**
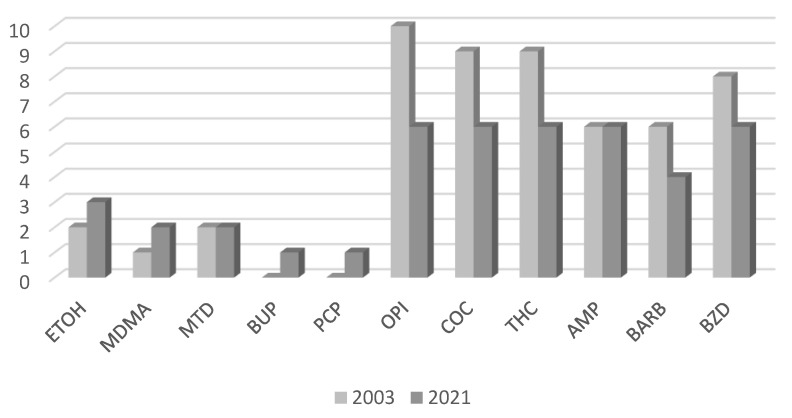
Total number of hospitals performing toxicological analyses for each substance of abuse. ETOH = ethanol; MDMA = 3,4-methylenedioxymethamphetamine (ecstasy); MTD = methadone; BUP = buprenorphine; PCP = phencyclidine; OPI = opiates; COC = cocaine; THC = tetrahydrocannabinol; AMP = amphetamines; BARB = barbital; BZD = benzodiazepines.

**Figure 6 pharmaceutics-15-02181-f006:**
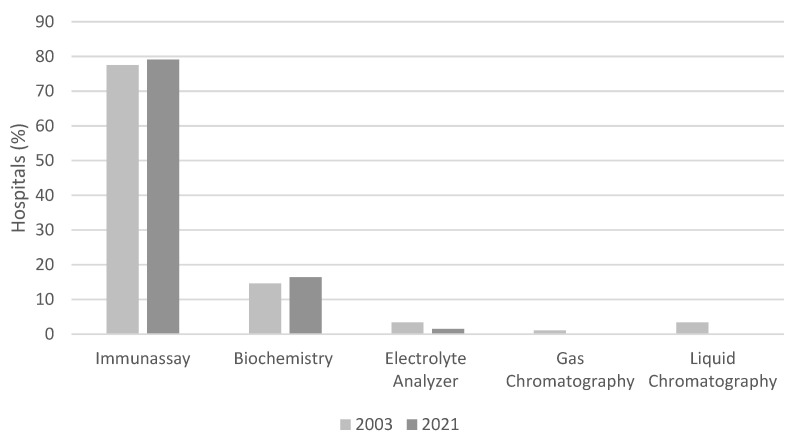
Percentage of hospitals performing TDM analysis using the indicated method in 2003 and 2021.

**Table 1 pharmaceutics-15-02181-t001:** Number of hospitals that performed TDM and toxicological analyses in 2003 and 2021.

	Year 2003	Year 2021
Greek hospitals included in analysis (n)	113	98
Performing TDM (n, %)	64 (56.6)	51 (52.0)
Greek hospitals included in analysis (n)	112	97
Performing Toxicological Analyses (n, %) Yes	9 (8)	6 (6.2)

TDM = therapeutic drug monitoring.

**Table 2 pharmaceutics-15-02181-t002:** Comparison between the numbers of hospitals that performed TDM measurements for each drug in 2003 and in 2021.

Drugs	2003 (n = 64)		2021 (n = 50)		*p*-Value *
	N	%	Ν	%	
Carbamazepine	38	59.4%	31	62.0%	0.265
Phenytoin	40	62.5%	28	56.0%	0.050 ^†^
Valproic Acid	39	60.9%	36	72.0%	0.728
Phenobarbital	34	53.1%	20	40.0%	0.016 ^†^
Lithium	14	21.9%	13	26.0%	0.999
Tricyclic Antidepressants	6	9.4%	2	4.0%	0.125
Aminoglycosides	3	4.7%	0	0	-
Amikacin	16	25.0%	8	16.0%	0.057
Gentamycin	12	18.8%	14	28.0%	0.815
Tobramycin	9	14.1%	2	4.0%	0.065
Vancomycin	21	32.8%	21	42.0%	-
Digoxin	52	81.3%	37	74.0%	0.038 ^†^
Cyclosporine	13	20.3%	13	26.0%	-
Tacrolimus	6	9.4%	10	20.0%	0.289
Theophylline	26	40.6%	7	14.0%	<0.001 ^‡^
Salicylate	14	21.9%	9	18.0%	0.302
Acetaminophen	10	15.6%	9	18.0%	-
Cortisol	5	7.8%	1	2.0%	0.125
Benzodiazepines Serum	8	12.5%	3	6%	0.125
Methotrexate	6	9.4%	8	16%	0.687
Insulin	2	3.1%	0	0	-
Primidone	10	15.6%	0	0	-
Topiramate	1	1.6%	0	0	-
Procainamide	1	1.6%	0	0	-
Netilmicin	9	14.1%	0	0	-
Everolimus	0	0	2	4%	-
Sirolimus	0	0	2	4%	-
Teicoplanin	1	1.6%	0	0	-
Levetiracetam	0	0	1	2.0%	-
Digitoxin	1	1.6%	0	0	-
Amiodarone	1	1.6%	0	0	-
Quinidine	1	1.6%	0	0	-
Ethosuximide	3	4.7%	0	0	-

* McNemar test in a total of 78 paired hospitals which performed at least one year or both years TDM. ^†^
*p* ≤ 0.05; ^‡^
*p* ≤ 0.001.

**Table 3 pharmaceutics-15-02181-t003:** The total and the mean number of TDM measurements per individual drug performed in Greek public hospitals in 2003 and in 2021.

	2003 (n = 64)		2021 (n = 50)		
	Total	Mean (SD)	Total	Mean (SD)	d (%)
Digoxin	38,439	600.6 (1080)	6440	128.8 (155.2)	−31,999 (83.2)
Valproic Acid	17,992	281.1 (431.5)	12,998	260.0 (342.5)	−4994 (27.8)
Cyclosporine	15,263	238.5 (856.1)	7500	150.0 (475.8)	−7763 (50.9)
Phenytoin	13,113	204.9 (334.3)	4709	94.2 (175.2)	−8404 (64.1)
Carbamazepine	11,374	177.7 (278.4)	2035	40.7 (53.5)	−9339 (82.1)
Tacrolimus	9470	148.0 (535.7)	7755	155.1 (711.5)	−1715 (18.1)
Amikacin	6314	98.7 (303.5)	3399	68.0 (354.7)	−2915 (46.2)
Phenobarbital	6292	98.3 (238.5)	1508	30.2 (57.3)	−4784 (76)
Theophylline	5276	82.4 (186.3)	419	8.4 (32.3)	−4857 (92.1)
Salicylate	3289	51.4 (209)	1058	21.2 (74.3)	−2231 (67.8)
Benzodiazepines	2690	42.0 (198.3)	646	12.9 (84.8)	−2044 (76)
Lithium	2506	39.2 (160)	1163	23.3 (57.2)	−1343 (53.6)
Methotrexate	1743	27.2 (114.9)	1407	28.1 (81.2)	−336 (19.3)
Tricyclic Antidepressants	1651	25.8 (141.2)	715	14.3 (71.3)	−936 (56.7)
Cortisol	1605	25.1 (115.8)	1	0	−1604 (99.9)
Netilmicin	916	14.3 (40.8)	0	0	−916 (100)
Tobramycin	915	14.3 (51)	63	1.3 (6.7)	−852 (93.1)
Primidone	419	6.5 (26.5)	0	0	−419 (100)
Aminoglycosides	270	4.2 (19.4)	0	0	−270 (100)
Amiodarone	140	2.2 (17.5)	0	0	−140 (100)
Ethosuximide	135	2.1 (12.8)	0	0	−135 (100)
Quinidine	100	1.6 (12.5)	0	0	−100 (100)
Procainamide	70	1.1 (8.8)	0	0	−70 (100)
Teicoplanin	30	0.5 (3.8)	0	0	−30 (100)
Digitoxin	30	0.5 (3.8)	0	0	−30 (100)
Topiramate	10	0.2 (1.3)	0	0	−10 (100)
Insulin	6	0.1 (0.6)	0	0	−6 (100)
Vancomycin	8507	132.9 (438.6)	29,459	589.2 (1433.8)	+20,952 (71.1)
Gentamycin	3997	62.5 (273.4)	6503	130.1 (332)	+2506 (38.5)
Acetaminophen	751	11.7 (36.1)	1145	22.9 (76.3)	+394 (34.4)
Everolimus	0	0	850	17.0 (87.8)	+850 (100)
Sirolimus	0	0	287	5.7 (30.6)	+287 (100)
Levetiracetam	0	0	5	0.1 (0.7)	+5 (100)

SD = standard deviation; d = difference (2021 − 2003).

**Table 4 pharmaceutics-15-02181-t004:** Comparison between drug measurements in the hospitals which performed TDM in both years.

Drugs	2003 (n = 35)	2021 (n = 35)	*p*-Value *	d (%)
Carbamazepine	198.8 (46.6)	46.6 (10.1)	<0.001 ^⁑^	−76.4
Phenytoin	253.6 (59)	120 (34.3)	0.001 ^⁑^	−52.8
Valproic Acid	315.2 (64.2)	313.3 (66.2)	0.981	−0.6
Phenobarbital	97.9 (30.8)	36.7 (11)	0.069	−62.2
Lithium	22.6 (11.1)	28.3 (11.1)	0.625	−25.2
Tricyclic Antidepressants	16.3 (10.2)	20.4 (14.3)	0.686	+25.2
Amikacin	147.3 (65.2)	91.1 (71.4)	0.033 ^†^	−38
Gentamycin	107.2 (61.6)	143.4 (61.2)	0.438	+33.8
Tobramycin	16 (7)	0.57 (0.6)	0.017 ^†^	−96.2
Vancomycin	206.1 (96.1)	789.1 (282.8)	0.012 ^†^	+282.9
Digoxin	783.2 (226.70)	165.9 (28.9)	<0.001 ^⁑^	−78.8
Cyclosporine	337.2 (170.3)	179.6 (92.6)	0.311	−46.6
Tacrolimus	122.3 (66.8)	214.2 (143.1)	0.484	+75.1
Theophylline	71.5 (28.7)	11.9 (6.4)	0.004 ^‡^	−83.3
Salicylate	45.7 (18.4)	29.3 (14.8)	0.414	−35.9
Acetaminophen	18.7 (7.9)	31.5 (15.2)	0.209	+68.5
Cortisol	41.4 (26)	0.03 (0.03)	0.068	−99.9
Benzodiazepines serum	34.9 (19.9)	17.9 (17.1)	0.249	−48.7
Methotrexate	29.2 (16.7)	37.7 (16.1)	0.735	+29.1

Data are shown as mean (standard deviation); d = difference (2021 − 2003), * Wilcoxon test. ^†^
*p* ≤ 0.05; ^‡^
*p* ≤ 0.01; ^⁑^
*p* ≤ 0.001

**Table 5 pharmaceutics-15-02181-t005:** Characteristics of laboratory type and laboratory staff where TDM is performed.

	2003	2021
	n = 113	n = 98
	N (%)	N (%)
Laboratory Type		
Biochemistry	42 (37.2)	40 (40.8)
Microbiology	13 (11.5)	4 (4.1)
Pharmacology	6 (5.3)	3 (3.1)
Biopathology	2 (1.8)	-
Toxicology	1 (0.9)	1 (1)
Immunology	-	3 (3.1)
Hormonology	1 (0.9)	
External Lab/Other Hospital	20 (17.7)	17 (17.3)
N/A	28 (24.8)	30 (30.6)
Laboratory staff ^a^	n = 242	n = 260
Physicians	69 (28.5)	56 (21.5)
Pharmacists	14 (5.8)	1 (0.4)
Chemists	30 (12.4)	44 (16.9)
Biologists	13 (5.4)	37 (14.2)
Biochemists	8 (3.3)	16 (6.15)
Lab Technicians	108 (44.6)	106 (61.5)

N/A = no answer. ^a^ n of hospitals responded 64 and 51 in 2003 and 2021, respectively.

## Data Availability

Data sharing not applicable.

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
