# Peer review of "Therapeutic Drug Monitoring (TDM) Implementation in Public Hospitals in Greece in 2003 and 2021: A Comparative Analysis of TDM Evolution over the Years"

_pharmaceutics, 2023, doi:10.3390/pharmaceutics15092181_

Round 1

Reviewer 1 Report

To the Authors

1.       Table 5 “laboratory staff”: please delete the term “doctors” and replace it with the term “physicians”. What is supposed to be a “phyisicist”?

2.       My main concern relates to the trend analysis dealing with the comparisons of TDM requests of single drugs between 2003 and 2021. These analyses are clearly biased by the different use of old drugs with time. For instance, it is expected that old antiepileptics have been replaced by new drugs and, consequently, the number of TDM requests are likely to be reduced…What about the TDM of new drugs? To strengthen the value of their research, the Authors should describe the overall TDM tests performed in 2003 and in 2021, including also the number and types of drugs monitored in the 2 periods.

Author Response

The point-by-point response is in the attached files. 

Reviewer 2 Report

The manuscript presents the results from TDM implemented in Greek hospitals in 2003 and 2021. Overall, I found the manuscript well-written and interesting. The main conclusion was that the number of TDM assays decreased in 2021 compared to 2003, which is very concerning considering the significant role that TDM plays in improving the efficacy and safety of pharmacotherapy. Thus, the results obtained by the Authors should be shared with the broader public to increase awareness of this disturbing trend. Below, please find a few additional comments that could help to improve the manuscript.

(1) The Authors should discuss their results in the context of the available literature. Was a similar trend observed in other countries?

(2) Why are the ‘study limitations’ mentioned at the end of ‘Discussion’ while the ‘study strengths’ are in the ‘Conclusions’? Also, the ‘Conclusions’ section should be shortened to summarize the obtained results better.

(3) Lines 86 and 91: Were these questionnaires sent in 2003 and 2021, or 2004 and 2022?

(4) Introduction and discussion: The Authors should pay attention to the used terminology (LC-MS/MS vs. LC/MS vs. LC-MS; also HPLC vs. UPLC – e.g., line 74 – UPLC can be combined with UV detector – the same as HPLC, or with a mass spectrometer, it does not exist as an independent instrument). These paragraphs (in the introduction and discussion) should be revised to ensure that correct information is provided.

(5) Line 101: Approval number should be provided.

(6) Table 1: Why are there different numbers of hospitals included in TDM and toxicological analyses?

(7) Line 161: Please, make sure there were 34 drugs (it does not corroborate with Table 2).

(8) Table 2: The percentages should be rounded to one decimal place for both 2003 and 2021. Was ‘cortisol’ included in TDM or a hormone level analysis? Why is ‘serum’ following benzodiazepines – does it mean all other drugs were determined in plasma?

(9) Figure 2: It is unclear why Fig. ‘c’ follows a different format than a, b, d, e; Fig. 2e – frequency cannot > 100%. If there were any statistically significant differences between 2003/2021 – it should be marked on the graphs.

(10) Line 244: Why 101? It does not comply with Table 1.

(11) Figure 6: % on the OY axis is missing.

The manuscript should be spell-checked for the correct use of English and punctuation (e.g., line 67 – period is missing at the end of the sentence; Table 2 -should be valproic ACID, tricyclic ACIDS; Fig. 2e and Fig. 3 – should be ANTIBIOTICS instead of antiobiotics; line 223 – ‘that’ should be removed; line 356 – S. aureus should be italicized, etc).

Author Response

(The authors gave the same response as above.)

Round 2

Reviewer 1 Report

No additional comments or requests

Author Response

Thank you very much for your initial comments, which helped us to improve our manuscript.  

Reviewer 2 Report

The manuscript has been extensively revised, and most issues have been resolved/explained. I have two additional comments:

1)     Lines 68 – 77 (and discussion) still need revision. What is the difference between LC-MS/MS (line 71) and HPLC-MS/MS (line 74) or UPLC-MS/MS (line 75) that the Authors mentioned as an additional technique? I would suggest not separating into LC/HPLC/UPLC. LC-MS/MS is a gold standard, and there is no doubt about it. This statement includes both HPLC-MS/MS and UPLC-MS/MS. The same with ultraviolet detection. 

2)     "No statistical analysis was applied to the data appeared in Figure 2.” In such a case, please perform statistical analysis to support the observations seen in the graph and include the statistics in the graph. The point of preparing such a graph is to show whether there was a difference between the years. And this information is meaningless without statistical analysis.

Author Response

Thank you very much for your comments.

You can find the responses to your comments in the attached file.
